# Thermal Reactivation of Hydrated Cement Paste: Properties and Impact on Cement Hydration

**DOI:** 10.3390/ma17112659

**Published:** 2024-05-31

**Authors:** Asghar Gholizadeh-Vayghan, Guillermo Meza Hernandez, Felicite Kingne Kingne, Jun Gu, Nicole Dilissen, Michael El Kadi, Tine Tysmans, Jef Vleugels, Hubert Rahier, Ruben Snellings

**Affiliations:** 1Vlaamse Instelling voor Technologisch Onderzoek, 2400 Mol, Belgium; a.gholizadeh@vandersanden.com (A.G.-V.); ruben.snellings@kuleuven.be (R.S.); 2Physical Chemistry and Polymer Science (FYSC), Sustainable Materials Engineering (SUME) Research Group, Vrije Universiteit Brussel (VUB), Pleinlaan 2, 1050 Brussels, Belgium; guillermo.meza.hernandez@vub.be (G.M.H.); kingnefelicite1@gmail.com (F.K.K.); jun.gu@vub.be (J.G.); nicole.dilissen@vub.be (N.D.); 3Department of Materials Engineering, KU Leuven, Kasteelpark Arenberg 44, 3001 Leuven, Belgium; jozef.vleugels@kuleuven.be; 4Department of Mechanics of Materials and Constructions (MeMC), Vrije Universiteit Brussel (VUB), Pleinlaan 2, 1050 Brussels, Belgium; michael.el.kadi@vub.be (M.E.K.); tine.tysmans@vub.be (T.T.); 5Department of Earth and Environmental Sciences, KU Leuven, Celestijnenlaan 200e, 3001 Leuven, Belgium

**Keywords:** recycled cement paste, thermal activation, reactivity, supplementary cementitious material, blended cement, hydration

## Abstract

In this research, the properties and cementitious performance of thermally activated cement pastes (referred to as DCPs) are investigated. Hydrated pastes prepared from Portland cement and slag blended cement were subjected to different thermal treatments: 350 °C for 2 h, 550 °C for 2 h, 550 °C for 24 h and 750 °C for 2 h. The properties and the reactivity as SCM of the DCPs were characterised as well as their effect on the mechanical performance and hydration of new blended cements incorporating the DCPs as supplementary cementitious materials (SCMs). It was observed that the temperature and duration of the thermal treatment increased the grindability and BET specific surface area of the DCP, as well as the formation of C_2_S phases and the reactivity as SCM. In contrast, the mechanical strength results for the blended cements indicated that thermal treatment at 350 °C for 2 h provided better performance. The hydration study results showed that highly reactive DCP interfered with the early hydration of the main clinker phases in Portland cement, leading to early setting and slow strength gain. The effect on blended cement hydration was most marked for binary Portland cement–DCP blends. In contrast, in the case of ternary slag cement–DCP blends the use of reactive DCP as SCM enabled to significantly increase early age strength.

## 1. Introduction

The production of Portland cement accounts for 5–8% of the total anthropogenic CO_2_ emission and consumes large amounts of natural resources [1,2]. On average, 842 kg of CO_2_ is generated by producing one ton of clinker [3]. The reduction in the use of clinker in new cement via supplementary cementitious materials (SCMs) is known to be one of the most effective ways of lowering the CO_2_ footprint of the cement and concrete industries. An interesting way of reducing clinker use is its partial replacement by recycled concrete fines (RCF) in new cement. RCF as one of the main concrete recycling products is usually not recommended for use as fine aggregate in new concrete and has thus far received relatively limited attention [4]. Several solutions have been proposed for the utilisation of RCF as a cementitious material. The use of RCF as a raw feed for the production of clinker is one such approach. The main challenges of this approach that limit high level incorporation are the high SiO_2_ content compared to Portland clinker raw meal, the presence of impurities and variations in the chemical composition of RCF. Compositional variability reflects the wide ranges in the chemical composition of cements and concrete aggregates and the varying ability of concrete recycling processes to liberate aggregates from hardened cement paste [5]. For instance, Gastaldi et al. [6] investigated the use of recycled concrete powder as a raw feed for clinker production and concluded that due to its composition, RCF content had to be limited to 20–40%. The direct use of milled RCF as SCM is attracting recent attention [7,8,9]. This approach offers more robustness for implementation on industrial scale. As the cement in RCF has largely reacted, RCF is considered to mainly act as a filler material when used as is [10,11]. Thermal treatment is reported to partially reactivate the hydrated cement paste contained in RCF providing some cementitious properties to this material [12,13,14,15,16]. 

Thermal treatment leads first to the gradual dehydration of different hydrates in the cement paste and, eventually, to the formation of new high-temperature phases with potential hydraulic activity. The degree of dehydration of each phase and the new phases formed are reported to depend on the treatment conditions such as temperature and time. The heating rate, oven/furnace type, particle size and cooling rate also affect the extent of dehydration and the phases formed [17,18]. Additionally, the type and composition of the initial cement play a key role in the dehydration behavior and the phase composition of the dehydrated cement product (DCP) [19]. A review by Carriço et al. [18] concluded that the literature shows some ambiguity regarding the phases formed during thermal activation of hydrated Portland cement. Formation of α’-, β- and γ-C_2_S has been reported at temperatures ranging from 600 °C to 900 °C by different authors, as measured with X-ray diffraction. Below 600 °C, a continuous dehydration of the interlayer water and gel water in C S H is reported, measured with TGA [20,21]. At around 750 °C, quantitative XRD showed poorly crystalline wollastonite (CS) and larnite (β-C_2_S), reported by Wang et al. [12]. Serpell and Zunino [17] reported quantitative XRD measurements of a decreasing α’- and γ-C_2_S content with increase in temperature at the expense of an increasing β-C_2_S content over the 600–800 °C temperature range.

It is important to determine the optimum dehydration temperature to achieve the highest degree of reactivity and strength performance. Recent studies mainly considered activation temperatures between 300–800 °C [22,23,24,25]. Wang et al. [12] found the highest compressive strength at ~450 °C when DCP is used as binder on its own. Angulo et al. [26] reported a comparable optimum dehydration temperature (500 °C). On the other hand, Shui et al. [27] found a direct correlation between increasing the activation temperature and the water demand and rate of setting of blended cements incorporating DCP. Improvement in the cementitious performance of DCP up to a dehydration temperature of 800 °C was reported, followed by a decline above 800 °C [27].

Somewhat different conclusions were reached for blended cements where DCP partially substitutes Portland cement as SCM. Florea [24] investigated the effects of using thermally activated RCF as SCM on the strength of mortar mixtures. It was concluded that a very fine fraction of recycled concrete treated at 800 °C can replace Portland cement up to 20 wt% without significant loss of strength. Wu et al. [28] investigated the effects of thermal treatment on the properties and cementitious performance of waste concrete, mortar and paste powder. They observed a similar behavior for all three materials concluding improvement in the performance of thermally treated waste in comparison with untreated waste when the dehydration temperature ranged from 600 °C to 1000 °C. An improvement in the 28-d strength activity index of waste paste powder from 70% to 90% was reported upon application of a thermal dehydration at 600 °C (when compared to the untreated powder). Inferior performance (compared to untreated powder) was however observed when the dehydration temperature was as high as 1200 °C.

There seems to be no consensus regarding the proper dehydration conditions when it comes to the strength performance and the phases formed during dehydration and rehydration. While the literature is generally in agreement regarding the high early heat of hydration of thermally activated cement, the impact of the activation conditions on the cement hydration reactions in blends with Portland cement is still controversial [18].

The objective of this paper is to obtain a deeper understanding of the influence of thermal treatment on the properties and cementitious performance of hydrated cement paste (as a proxy to recycled concrete fines). It is hypothesized that the thermal treatment of hydrated cement paste, CEM I and CEM III, could significantly enhance the performance of OPC as a supplementary cementitious material (SCM). To this end, a systematic study is carried out on the effects of different thermal treatments on the physical, chemical and mineralogical characteristics of hydrated cement paste (produced from Portland cement (CEM I) and slag cement (CEM III)). The impacts of using such materials as SCM on the hydration process and strength development of blended cements is investigated through a comprehensive hydration study, and the underlying root causes for the observed effects are explored and interpreted. The cementitious performance of dehydrated cement paste (DCP) produced under different conditions are also compared, and the most suitable treatment conditions are identified for both CEM I and CEM III cases.

## 2. Materials and Methods

### 2.1. Materials Selection and Preparation

Commercial cements, CEM I and CEM III/A (prepared by blending 300 kg of CEM I with 220 kg of ground granulated blast furnace slag, or GGBFS), were used to prepare the initial hydrated cement pastes (later to be dehydrated and studied). Table 1 shows the chemical, physical and mineralogical properties of the CEM I, GGBFS and CEM III used in this research. A fine quartz filler was also used as a reference inert material to be compared with the thermally activated dehydrated cement paste (DCP) fines. Figure 1 shows the particle size distributions and the characteristic particle sizes of the above materials. Particle size distribution was analysed using a Horiba laser diffraction particle size analyser in isopropyl alcohol (IPA).

For the preparation of the dehydrated cement fines, cement pastes with a water-to-binder ratio of 0.5 were first prepared from both CEM I and CEM III. The pastes were demoulded after 1 day and cured in saturated lime water until 28 days at 21 ± 2 °C. Next, the hydrated cement pastes were crushed to below 20 mm and dried at 40 °C for 48 h. The dried materials were then introduced into a muffle furnace and heated at a rate of 10 °C/min up to the target dwell temperatures of (1) 350 °C for 2 h, (2) 550 °C for 2 h, (3) 550 °C for 24 h and (4) 750 °C for 2 h. Dwell times of 2 h were selected to ensure homogeneous heating of the materials, while a dwell time of 24 h at 550 °C was used to investigate the effect of heating time on the reactivity of DCP and its influence on the mechanical properties. Upon completion of the thermal cycle, the materials were removed from the furnace and allowed to cool down to ~170 °C (temperature monitored via an IR-pyrometer) and subsequently transferred to a desiccator. A planetary ball mill with a tungsten carbide jar (500 mL) and 13 × 2-mm tungsten carbide milling balls was used for milling the materials. A 60 g quantity of dried DCP (added with 5 wt% of isopropyl alcohol as the grinding aid) was milled at 350 rpm for 12 min (with 1 min inversion intervals) in each cycle. The milled products were then dried at 60 °C for 24 h, fully blended and characterised for their physicochemical properties and used for preparing new blended cement paste and mortars for further studies. Table 2 shows an overview of the DCPs used in this study along with their labels.

### 2.2. Characterisation Methods

#### 2.2.1. Material Characterisation

The particle size distributions of the milled DCPs were analysed using a Horiba laser diffraction particle size analyser in IPA. Ultrasonication was applied to disperse the particle agglomerates in another IPA solution prior to measurement. The reported results are the average of three measurements. The fineness of the DCP powders was measured via the Brunauer-Emmett-Teller (BET) nitrogen gas sorption method using a Autosorb iQ Series instrument (Anton Paar QuantaTec Inc., Boynton Beach, FL, USA).

The DCPs were subjected to scanning electron microscopy to observe the effects the morphology of the DCP particles after milling. Secondary electron images were acquired using a FEI FEG Nova NanoSEM 450 (Thermo Fisher Scientific, Waltham, MA, USA) operated at an electron acceleration voltage of 5 kV.

The bound water, portlandite and carbonate contents of the DCPs were measured via thermogravimetry. Samples were heated at 10 °C/min from 30 °C up to 1000 °C under nitrogen atmosphere at a 50 mL/min N_2_ flow rate. The bound water was determined by measuring the mass change between 50 °C and 550 °C and the portlandite and carbonate contents were estimated respectively as the mass loss in the 400–500 °C and 600–790 °C ranges using the tangential method [29].

The phase composition of the DCPs was determined via X-ray diffraction. XRD data were collected using an Empyrean diffractometer (Malvern Panalytical Ltd., Malvern, UK) equipped with a CoKα tube operated at 40 kV and 45 mA. Diffraction scans were recorded from 5 to 110° 2θ, with step size 0.013° 2θ and a measurement time of 50 s per step. Rietveld analysis was performed using HighScore Plus software (version 4.6a) and the external standard approach was used to determine the amorphous phase content (rutile; Kronos 2300 TiO_2_ external standard calibrated against NIST SRM 676a α-Al_2_O_3_). The starting crystal structure data were taken from Snellings [30] and the chemical composition of the DCPs (cf. Table 1) was used to calculate the mass attenuation coefficients needed for the absorption correction.

The chemical reactivity of the DCP powders was measured using the R3 (rapid, relevant, reliable) reactivity test according to ASTM C1897:2020 [31]. Test mixtures were produced incorporating the DCPs and Ca(OH)_2_, H_2_O, with additions of CaCO_3_, K_2_SO_4_ and KOH to simulate the reaction environment of a hydrating Portland cement. A freshly made 15.0 g paste specimen was prepared, and the cumulative heat was recorded.

#### 2.2.2. Hydration Study

A hydration study was carried out on the DCPs in comparison with the selected inert filler as follows. Cement pastes incorporating DCPs were tested for their phase evolution (via X-ray diffractometry), hydration kinetics (isothermal calorimetry) and thermogravimetric response (TGA). The control specimens were 100% CEM I or CEM III pastes with a water-to-binder ratio of 0.40. The blended mixtures were produced by replacing 30 wt% of the cement with either DCP of the same type. The mixtures are labelled based on the type of the cement used and the replacement type. I-350/2 for instance refers to the paste made with CEM I and 30 wt% replacement with DCI/350 °C/2 h. A mixture incorporating 30 wt% cement replacement with quartz powder (I-QP and III-QP) as inert filler was also produced to account for the filler effect of the DCPs. All pastes were prepared by first dry-mixing the powders for 60 min using a Turbula multidirectional mixer. The dry mixture of the main ingredients was then mixed with water at 1600 ± 50 rpm for 2 min using a high shear blender so that a homogeneous paste was achieved.

All prepared cement pastes were exposed to extensive XRD investigations as follows. The early-age behaviour of all CEM I paste series were explored using in situ XRD in 15-min time intervals (started within 15 min after mixing) up to 24 h. The fresh pastes were cast into XRD sample holders and covered with a semi-spherical X-ray transmissible polymer dome to avoid surface evaporation. The closed sample holder was placed inside the XRD measurement position for continuous data collection during early hydration. Diffraction scans were recorded from 5 to 55° 2θ under the conditions mentioned above. This was done to monitor the early age formation of hydrated phases and allow for the comparison of the effects of incorporating different DCPs in the cement pastes. The longer term evolution of hydration phases was also studied via ex situ XRD from 1 d to 91 d for both CEM I and CEM III pastes (in the 5 to 90° 2θ range). To this end, the prepared cement pastes were cast into 12-mL plastic moulds and stored at 21 ± 2 °C for 24 h. Next, the hardened pastes were demoulded and stored inside slightly larger plastic bottles under a thin film of water until the testing age. At each testing age, the hardened pastes were taken out, a slice was cut, and the surface of the freshly cut specimen was smoothened using a 400 grit sandpaper and then flushed with water. The extra water film was removed from the surface, and the specimen was directly subjected to the test under the same conditions as mentioned above.

The thermogravimetric response of the hydrated cement pastes was measured in order to determine and compare the bound water and portlandite contents of the control and blended pastes over time. Representative hydration-stopped samples were heated at 10 °C/min from 30 °C up to 1000 °C under nitrogen atmosphere at 50 mL/min flow rate. The bound water was determined by measuring the mass change between 50 °C and 550 °C and the portlandite content was estimated as the mass loss in the 400–500 °C range using the tangential method. The values are reported with respect to the residual masses at 550 °C to cancel out the dilution effects due to the incorporation of water.

A sample of each paste mixture was prepared to determine the heat of hydration using isothermal calorimetry. The heat evolution was then monitored at 20 °C using a TAM Air calorimeter up to 7 days according to the Method B of ASTM C1702 [32].

#### 2.2.3. Mechanical Performance

To assess the cementitious performance of the produced DCPs, standard mortar mixtures conforming to EN 196-1 [33] specifications were produced and tested at different ages for their compressive strength. A water-to-binder ratio of 0.5 and a sand-to-binder ratio of 3.0 were used for all mixtures. The binder in the control mixtures was composed solely of either CEM I or CEM III. In the case of test (blended) mixtures, 30 wt% replacement with each DCP was made in order to observe the influence of such replacements on the water demand and strength development of the mortars. For each cement type, DCPs originally produced from the same cement type were used as partial replacements. The mixture compositions are the same as those of the cement pastes. For each composition, polycarboxylate-based high-range water reducing admixture (also referred to as superplasticizer: SP) was added to the mixing water in order to maintain the mortar flow in the 175 ± 15 mm range. The SP addition ranged between 0.04 and 0.22 wt.%, increasing with DCP treatment temperature. This approach was taken to eliminate the effects of DCPs on the consistence and compactability of the mixtures and their subsequent effects on the mechanical properties of the hardened mortars. Mixtures with a flowability outside the designated ranges were disposed and new mixtures with adjusted superplasticizer contents were produced. Once the flowability in the desired range was obtained, the mixture was cast into three 40 × 40 × 160 mm^3^ prismatic moulds, covered with plastic sheets and stored at 20 ± 2 °C for 24 h. Each mixture was produced four times (one batch for each testing age: 1, 7, 28 and 91 d). The hardened specimens were removed and cured in a climate chamber at 20 ± 1 °C and RH > 95% until the time of testing.

## 3. Results and Discussion

### 3.1. Material Characterisation

Figure 2 shows the particle size analysis results of the dehydrated cement paste powders along with the summarised characteristic particle sizes. It could be observed that the milled materials are in all accounts of similar size to the respective starting cement or finer (compare the characteristic particles sizes shown in Figure 2 with respective values in Figure 1). Comparing the particle size of DCPs produced under different treatment conditions, it could be concluded that an increase in the dehydration temperature and time generally leads to a higher grindability (note the order of the PSD curves in Figure 2 with respect to the colour codes). For both CEM I and CEM III, the DCPs produced at 350 °C have the largest average particle size and D90 (see the blue curves), whereas DC/550 °C/24 h and DC/750 °C/2 h show the smallest values (see the orange and red curves, respectively).

The BET specific surface area values of the DCI and DCIII powders are shown in Figure 3. For DCI, the BET surface monotonically increases with increase in the dehydration time and temperature. Similar behaviour was also observed in the case of DCIII with the exception of DCIII/750 °C/2 h. A considerable decline in the BET surface area of DCIII is observed when the dehydration temperature exceeds 550 °C. It could be argued that as the dehydration temperature increases, the C-S-H continues to dehydrate, shrinks and becomes more porous [34]. However, once the dehydration temperature exceeds a certain level, a major part of C-S-H recrystallises into C_2_S, which entails a collapse in the porous structure by nucleation and growth of C_2_S crystals. An important finding from the BET results is that the DCPs (and consequently RCF) have a very high specific surface area as a potential SCM. The BET surface areas observed in the DCPs’ range from 15 m^2^/g to 23 m^2^/g, which is more than one order of magnitude higher than those of the starting materials (i.e., CEM I and GGBFS: 1.76 and 1.35 m^2^/g, respectively). Such high porosity could be considered a drawback for DCP/RCF as it increases the water demand of the binder.

A closer look at the microstructure and morphology of the DCPs produced under different thermal conditions is provided in Figure 4 and Figure 5. It is observed that the DCP grains are irregularly shaped agglomerates consisting of flakes to globular sub-micron-sized particles. The DCPs produced at lower temperatures tend toward a more flake-like texture while those dehydrated at higher temperatures are more globular. Some agglomerates showed clear internal microporosity (e.g., in the case of DCIII/550 °C/2 h) (see Figure 5b), however this type of grains represented only a minority and was found for different treatment conditions. Therefore, this texture type is likely inherited from more porous zones in the original hydrated cement paste.

Figure 6a,b show the TGA responses of the DCI and DCIII materials, respectively. Using the TGA curves, the amount of bound water, portlandite content and carbonate content of DCI and DCIII were calculated and plotted in Figure 7a,b, respectively. The first remarkable difference is found in the bound water contents of DCPs prepared at 350 °C compared to those prepared at higher temperatures. With the increase in the dehydration temperature and time, the amounts of bound water and portlandite content decrease in all cases. It is interesting to note that the dehydration never led to a complete absence of portlandite, as would be expected for treatment temperatures of 550 °C and 750 °C. A shift to lower temperatures is observed for the dehydroxylation event of portlandite for 550 °C/24 h and 750 °C/2 h materials. This indicates a change in portlandite properties such as crystal size and thus may indicate a different origin; for instance, rapid hydration of free lime is known to result in small submicrometer crystal sizes [35]. Free lime formed by thermal decomposition of portlandite in cement pastes is known to be reactive and convert readily upon brief exposure to ambient air to either calcium hydroxide or calcium carbonates [36]. Minor mass losses were also observed at temperatures exceeding 600 °C, which indicates occasional formation of carbonates as a result of ambient air exposure during the handling steps.

The phase composition of the DCPs was studied in detail via XRD. The summarised Rietveld analysis results are presented in Figure 8a,b. It can be clearly observed in both figures that the thermal treatment causes major drops in the amorphous content of the DCPs. This is to the most part due to the formation of C_2_S phases and at 750 °C/2 h also C_4_AF and C_3_A. Various C_2_S phases could be refined; β-C_2_S is present after all treatments and increases in content with temperature. Below 550 °C, β-C_2_S is interpreted to be a residual clinker phase deriving from the initial CEM I and CEM III, and at 750 °C it appears to form significantly, despite being, as a pure phase, thermodynamically unstable. The more stable γ-C_2_S is the dominant C_2_S phase at 550 °C, yet becomes subordinate to β-C_2_S at 750 °C. No such phase was detected for DCP produced at 350 °C. While the α-C_2_S content appears to be within the error margin of ±2 wt% [37], it shows a notable increase at 750 °C in DCI. Although α-C_2_S is a high-temperature phase, its formation during HCP dehydration at low temperatures has been reported by other researchers as well [17,19]. The formation of high-temperature C_2_S polymorphs may be unexpected. In the case of α-C_2_S, the phase content refinement could be affected by misfitting of other phases of low crystallinity. However, β-C_2_S formation was clearly distinguished and identified at temperatures well below its established stability field. This apparent disparity may be explained by the observation that, in the absence of an efficient reaction medium such as a solution or a melt phase, the thermally induced solid-state conversions may be controlled by kinetic factors such as precursor or templating effects, as seen in the transformation of meta-clays from clay minerals [38]. Such effects have not been explored extensively for C-S-H in actual cement pastes as yet. In addition, the presence of impurities and defects in C-S-H may induce the formation of metastable phases and preserve these during cooling [39]. The C_3_S identified in the DCPs is interpreted to be a remainder of the initial cements. The variations in the C_3_S content are within the error range of the analysis. Some increase in the C_3_A and C_4_AF contents was observed at 750 °C, which is indicative of initial recrystallisation at such temperature. Moreover, it was noticed that the AFm and AFt phases were decomposed for the most part during thermal treatment at temperatures as low as 350 °C. No traces of such phases were observed at higher temperatures, where the presence of anhydrite indicates that such phases were at least partially decomposed and not merely dehydrated.

The changes in the phase compositions of the DCPs was observed to entail variations in their chemical reactivity. Depending on the treatment conditions, DCI and DCIII showed 7-d R3 reactivities in the ranges of 125–210 J/g and 225–283 J/g, respectively (see Table 3). The R3 heat release and heat flow results of DCI and DCIII materials are shown in Figure 9a,b. Notice the differences in the heat release values of DCPs produced under different conditions. Dehydration at 350 °C yields the least reactive DCPs in both cases. DCI produced at such temperature shows 45% lower reactivity compared to the DCPs obtained at higher temperatures and DCIII shows 25% lower reactivity.

DCPs produced at 550 and 750 °C show more or less similar reactivities. An attempt was made to find correlations between the phase composition of the DCPs and their R3 heat release so as to determine which phases potentially contribute to the generation of heat. The analyses indicate that while all high-temperature phases have positive effects on the R3 heat release, no clear statistically significant correlation could be detected between such phases and the 1-, 3- and 7-d R3 heat results. Table 4 lists the linear correlation coefficients between the R3 heat results (along with the corresponding *p*-values) and different mineral phases in DCI and DCIII. It is likely that the dynamics and the evolving characteristics (composition, crystallinity, defect density, etc.) of all phases contained in the DCPs are interfering with a straightforward correlation analysis. In particular, the properties of the large amorphous fraction would be significantly different for the different thermal treatments. Nevertheless, it is clearly observed that thermal treatment at 550 °C and above lead to an increase in the chemical reactivity of the DCPs. It should also be noted that the heat release values recorded for DCIII are greater than those of DCI in all cases in spite of DCIII having lower clinker, β-C_2_S and portlandite contents. This is most likely due to the chemical reactivity of the residual (i.e., unhydrated) GGBFS in DCIII.

### 3.2. Hydration Study

The effect of 30 wt% replacement of the Portland cements (CEM I and CEM III) by their respective DCPs on the hydration processes and the strength development was studied in a hydration study comprising isothermal calorimetry and thermogravimetric and XRD analysis. This was carried out with respect to a control cement of 100 wt% original CEM I and CEM III (I-CTRL and III-CTRL, respectively) and the non-reactive 30 wt% replacement by quartz powder (I-QP and III-QP, respectively). The calorimetry results (cumulative heat and heat flow) of the CEM I pastes are plotted in Figure 10. It is observed that all mixtures containing DCPs have a significantly higher heat flow in the initial hydration period (<1.5 h) and a shorter induction period compared to I-CTRL and I-QP. Such increase in the instant heat release and the shortening of the induction period is more significant in the case of DCI/550 °C/24 h and DCI/750 °C/24 h. Incorporation of all DCPs also leads to an acceleration of the main hydration peak, as indicated by the steeper slope of the heat flow curve during the acceleration stage and the earlier occurrence of the maximum heat flow.

With increase in the dehydration temperature and time, the heat flow peak occurs earlier, yet the area under the first peak decreases. Simultaneously, the main hydration peak is observed to split in two, and a second, broad shoulder or peak is observed at later hours (peak at 12–14 h); the second peak is clearly evident in I-750/2. Specifically in the case of this paste (i.e., I-750/2) and I-550/24 to some extent, a significant deviation in the heat release pattern from that of the control mixture is evident. Judging by the R3 results (Figure 9), it appears that the more reactive the DCPs, the larger the deviation is the heat flow pattern from that of the control paste. It can therefore be argued that the rapid hydration of the DCPs interferes with the hydration of the Portland clinker phases of the CEM I. Since the main hydration peak period is generally associated with the hydration of C_3_S [40], the very early-age C_3_S hydration is investigated later on using in situ XRD.

The incorporation of DCPs in CEM III pastes leads to similar effects as described for CEM I. In Figure 11, the heat flow and cumulative heat data are presented for the CEM III pastes. It can be observed that the split of the main hydration peak into an accelerated early peak and a retarded second peak is even more pronounced than in the CEM I pastes. In the case of the III-550/24 and III-750/2, the early peak has shifted to such an extent that it coincides with the initial dissolution peak. The large peak observed in the case of III-QP is indicative of sulfate depletion (due to the dilution effect), which is because CEM III was prepared merely by combining CEM I and GGBFS without the addition of calcium sulfates.

Regarding the main pastes, similar heat evolution patterns as those of DCI-blended pastes could be identified for the DCIII-blended pastes with a few remarks. The instant heat release observed in the pastes incorporating DCIII is significantly larger than that of respective CEM I pastes and large heat flows were detected for CEM III pastes incorporating DCIII dehydrated at 550 °C and above (see Figure 11). The shift of the acceleration peak to the left (compared to the control) is also more severe when comparing each blended CEM III paste with the respective paste in the CEM I series (Figure 10). While III-350/2 and III-550/2 show an initial peak at early hours, III-550/2 also shows a secondary peak at ~10 h which was not the case for I-550/2. III-350/2 also shows a minor secondary peak at 15–17 h. No initial peak in the heat flow of III-550/24 and III-750/2 could be identified while similar to III-550/2, a secondary peak around ~10 h could be observed for these pastes. All pastes incorporating DCIII exhibit higher heat of hydration compared to the control at all times while falling short from that of III-QP between 8 and 15 h. At later times, pastes incorporating DCIII show higher heat compared to both III-CTRL and III-QP and those incorporating DCPs produced at higher dehydration temperatures and times generate higher heats.

The modification of the heat release pattern in the first 48 h does not translate to lower cumulative heat in the longer term. By 7 days, the 550/24 and 750/2 DCPs for both CEM I and CEM III show the highest heat release, as normalised to the initial CEM I content (Figure 10b). On the other side of the spectrum, I-350/2 and III-350/2 show the least deviation from the heat evolution of I-CTRL and the lowest total heat release among all blended mixtures (and lowest R3 heat release both in the short and long terms). A good correlation could be established between the long-term heat release of the pastes incorporating each DCP and the R3 cumulative heat release of the same DCPs. In Figure 12, the isothermal calorimetry results of CEM I and CEM III pastes are plotted against the R3 heat results of the same at different test durations. Notice the direct correlation between the two.

Further investigations are carried out using thermogravimetric and XRD analysis to establish the effect of the DCPs on the formation and properties of the hydration product assemblages. Figure 13 and Figure 14 show the TG and DTG plots of CEM I and CEM III pastes for different hydration ages, respectively. The pastes incorporating DCPs show higher mass loss than the pastes with quartz powder at all ages. This indicates a higher content of hydrates contributed by (re)hydration of the DCPs. Compared to the control pastes (I-CTRL and III-CTRL), similar TG/DTG responses have been recorded for the pastes incorporating DCPs, indicating that the reaction of the DCPs produces similar types and amounts of hydration products. Part of these products are initially contributed by the DCPs. The bound water and portlandite contents of the blended pastes produced with and without DCP were calculated from the TGA experiments, presented in Figure 15 and Figure 16. Figure 15a confirms that the blends with quartz powder (I-QP) contain significantly less bound water than the I-CTRL and the DCP pastes, which is explained by the dilution of the cement by the inert quartz powder. The control and the DCP blends show similar levels of bound water.

Figure 16a shows the bound water variations of different CEM III pastes over time. In comparison with CEM I pastes, it is clearly observed that the CEM III pastes have a slower rate of hydration (slower rate of bound water generation). As in the CEM I mixtures, III-QP produced significantly less bound water compared to III-CTRL and the DCP blends at all ages. The bound water levels in the DCP blends were comparable to those in the III-CTRL paste, clearly indicating the contribution of the rehydrated DCPs to the overall hydrate content. Contrary to the increasing trend of the portlandite content with age in the CEM I mixtures, the portlandite content in the CEM III mixtures decreases over time in all cases. This is interpreted to be due to the presence of slag which consumes the portlandite. The lower portlandite contents in the III-QP pastes compared to III-CTRL can be accounted for by the 30 wt% dilution factor of the III-QP blend. In contrast, the lower portlandite levels in the DCIII pastes could indicate a higher overall degree of reaction of the slag, partially inherited from the DCIII materials, or, alternatively, a difference in the clinker hydration degree.

The consumption of the clinker phases and the formation of the hydration products was studied in more detail by quantitative XRD analysis. To identify the reaction mechanisms underlying the deviant early age heat release profiles of the DCP pastes, CEM I pastes were closely investigated in terms of their early age phase evolution via in situ XRD. As an example, Figure 17a and b show the evolution of phases in the I-350/2 paste in the 8–15° 2θ and 33–41° 2θ ranges, respectively. The diffractograms presented in Figure 17a,b were deliberately chosen to clearly show the evolution of the phases present in the samples. Figure 17a shows the formation of ettringite and hemicarboaluminate in the early hours of hydration. Figure 17b illustrates the consumption of the clinker phases (A and B; alite and belite) over time, giving rise to the formation of portlandite up to 24 h of hydration. Later age specimens were investigated by ex situ measurements on cut slices of hydrated paste. For illustration, the diffractograms of the I-350/2 and III-350/2 pastes for different hydration ages are presented in Figure 18. The longer term hydration of the DCP-cements shows broad similarities with the hydration of a reference Portland cement. Ettringite and portlandite form early on, while the AFm phases (hemi- and monocarboaluminate) form subsequently. In the CEM III blended cements, hydrotalcite is formed additionally as a reaction product of the hydration of the slag. A comparison of the hydrate assemblage of the various cements at the age of 28 days is made in Figure 19. Clearly, the DCP-blended pastes show higher, more pronounced ettringite and AFm (hemi- and monocarboaluminate) reflections than the control pastes.

The XRD data were processed by Rietveld analysis for quantification of phase abundancies in the course of hydration. Figure 20 presents the quantification results for both in situ and ex situ XRD measurements of the CEM I pastes. Inspection of the C_3_S degrees of reaction in Figure 20a,b reveals that at a very early age, C_3_S hydration initiates more early on in the DCP, yet is surpassed within several hours by the control paste as the C_3_S hydration rate in the DCP pastes appears slower after the first few hours. In line with the calorimetry results the rate of reaction of the C_3_S in I-CTRL slows down strongly after 7 h of hydration, allowing the DCP pastes to gradually catch up over the first few days of hydration, reaching equivalent levels by 2 days of hydration and beyond. The higher the dehydration temperature and duration of the DCP, the more pronounced the difference in the C_3_S reaction profile with the control.

Considering the total degree of hydration of clinker phases, Figure 20c,d present generally slower and lower degrees of reaction that persist until 28 days for DCPs dehydrated at temperatures of 550 °C and above. It should be noted that the C_2_S, C_3_A and C_4_AF phases, including their polymorphs, contained in the DCPs were integrated into the total number of clinker phases, as it was not possible to make a distinction between these and corresponding phases present in the CEM I. Therefore, to some extent, the lower degrees of reaction of the DCP containing pastes can be allocated to a lower reactivity of part of the DCP “clinker” phases, in particular the γ-C_2_S. The absence of such recrystallised clinker phases in I-350/2 can also explain the higher clinker degrees of reaction in the corresponding pastes. The early age portlandite content profiles in Figure 20e) mainly reflect the C_3_S reaction curves when taking into account that the DCP pastes initially contained portlandite. As shown in Figure 20f, at a later age, total portlandite levels in the DCP pastes appear to be slightly lower or equivalent to the control paste, which is more or less in line with the trend in the TGA results.

The XRD analysis results in Figure 20g,h indicate more rapid and more extensive formation of ettringite in the cements blended with DCPs, in particular for DCPs treated at higher temperatures. Also, the formation of AFm-hemicarboaluminate occurs significantly earlier in the blends with the higher temperature DCPs. This is a striking difference with the I-CTRL paste, in which hemicarboaluminate was not observed in the in situ experiment and was only found after 7 days of hydration in the ex situ measurements. Simultaneously, gypsum in the DCP cements is very rapidly consumed between stages, leaving no detectable traces in early hours (<1 h), while persisting until 4–5 h of hydration (beyond the main hydration peak) in the control paste. See Figure 21a,b to compare the gypsum primary peak between the I-CTRL paste and I-350/2 in early hours.

Rapid gypsum depletion and early hemicarboaluminate formation are both indicative for undersulfation of the DCP cements. Undersulfation is well known to affect the hydration of C_3_S, the nucleation and growth of C-S-H [41,42] and reduced strength development [43]. In the case sulfate depletion occurs before the main hydration peak, the C_3_S hydration peak becomes lower and broader [44]. It is argued here that the addition of DCP as a high surface area, reactive material perturbs the regular C_3_S hydration process by rapidly consuming sulfates from the system, either by precipitation of calcium aluminate hydrates, as observed by XRD, or by sorption onto C-S-H or DCP phases [45]. In addition, it can be argued that the DCP hydrates could act as suitable nuclei for new hydrates and in this way shorten the induction period between the initial peak and the main C_3_S hydration peak significantly. This is clearly observed in the calorimetry curves in Figure 10 and Figure 11 and to a lesser extent in the XRD results in Figure 20 by the earlier initiation of C_3_S hydration in the DCP pastes. A similar effect is sorted by commercial additives using homogeneously dispersed C-S-H nuclei to accelerate the hydration of C_3_S [46], however without the occurrence of the subsequent retardation seen by this study. The in situ XRD results indicate that this retardation is reflected in simultaneous slow-down in C_3_S hydration; however, the mechanism explaining this retardation is still unclear. Possible causes are one or a combination of the following: (i) modification of the pore solution composition, (ii) changes in the nucleation and growth rate of the C-S-H and other hydrates and (iii) differences in the resulting microstructure of the hydrated cement. The latter could entail changes in the mechanical performance of the cement which are presented and discussed in the subsequent section.

### 3.3. Mechanical Performance

The mechanical strength results indicate that the modification of the hydration process by introduction of the DCP phases can strongly affect the mechanical performance. Figure 22a,b represent the absolute and relative strength development results of the CEM I mortars. It is observed that I-550/24 and I-750/2 developed the lowest strength values among other mixtures containing DCP. They also showed lower strength values compared to I-QP. The inverse influence of incorporating such DCPs on the strength of CEM I mixtures is beyond their dilution effect and even indicative of some non-compatibility with CEM I in terms of strength development. I-550/2 and I-350/2, on the other hand, develop a 28-d compressive strength of nearly 70% that of the control and continue to close the gap at later ages. Therefore, they could be deemed moderately reactive SCMs.

A close look at the strength development plots of the CEM III mortar series (Figure 23) reveals that the presence of slag causes some notable changes in strength development. The gap between the strength performance of the DCP-blended pastes compared to the control is much smaller in such blends compared to the CEM I paste series. All DCP-blended mixtures have performed better compared to the mixture III-QP incorporating the inert filler (quartz powder) almost at all ages. III-550/24 was found to develop the highest strength values compared to all other DCP-blended mixtures at all ages and even better than the control mortar III-QP at early ages (<7 d). This mixture was also found to develop a 56-d strength, exceeding the 28-d strength of the control mixtures. While the rate of strength development is slower in the CEM III mixtures compared to CEM I mixtures (compare Figure 22b and Figure 23b), the CEM III mixtures reach higher strength values at later ages in comparison with their respective CEM I mixtures. Such systematic behaviour suggests that recycling of concrete produced from CEM III is technically more favourable when it comes to mechanical performance.

The phase composition and strength performance of DCPs are further analysed with respect to one another to explore correlations between the two. No positive correlation was detected between the belite phases formed in DCI/DCIII and their strength performance (see Figure 8, Figure 22 and Figure 23). DCI/750 °C/2 h, for instance, contained the highest β-C_2_S (see Figure 8) and showed the lowest strength. In addition, it is observed that there is a negative correlation between the compressive strength of the CEM I mixtures containing DCPs and the amounts of bound water and portlandite produced in the same. This observation coupled with the inverse correlation observed between compressive strength and heat release of DCP-blended mixtures (Figure 24) suggests that the hydrates contributed by the DCPs are apparently not favourable in terms of mechanical strength. DCI/350 °C/2 h, which generated the lowest R3 heat and which had lowest bound water and portlandite content, showed the highest compressive strength results compared to other blended mixtures at all ages.

Similar to the CEM I mixtures, the CEM III mixtures containing DCPs showed an inverse correlation between the compressive strength and the bound water and portlandite content. It appears that the DCP hydration products do not contribute to higher mechanical strength. The contribution of GGBFS, on the other hand, has positive effects on strength. Judging by the decreasing portlandite contents of CEM III pastes incorporating DCP, it could be argued that more GGBFS reactions translate to higher compressive strength (see III-350/2 and III-550/24 in Figure 16 in comparison to other mixtures).

## 4. Conclusions

In this research, dehydrated cement (DC) powders (referred to as DCPs) were prepared via thermal treatment and comminution of CEM I and (synthetic) CEM III hydrated pastes to investigate their physico-chemical properties and cementitious performance as supplementary cementitious materials (SCMs). A full hydration study was conducted to explore the effects of partially replacing fresh cement (both CEM I and CEM III) with such SCMs. The hypothesis that DCPs can be used as SCMs was only confirmed in part, and care should be taken because DCPs change workability, reactivity and final strength. The main findings of this research are summarized as follows.

Increasing the dehydration temperature from 350 °C to 750 °C generally results in an increase in the grindability of these materials and their surface area, with an exception observed for DCIII dehydrated at 750 °C. Recrystallization of C-S-H at 550 °C and higher temperatures leads to the formation of γ-C2S and β-C2S. The content of potentially hydraulic phases increased with dehydration temperature at the expense of the amorphous or nanocrystalline phase (likely C-S-H and other cement hydrates).

According to the R3 reactivity results, DCPs prepared at higher dehydration temperatures and times generate significantly higher initial rates of reaction compared to those prepared at lower temperatures and also show higher reactivity over the long term (after 1–2 days). Positive correlations were found between the R3 heat results and the amount of potentially hydraulic phases formed in the DCPs (for both DCI and DCIII), although most correlations were not statistically significant. However, no direct correlation was found between the R3 heat release and the strength performance of blended cements comprising 30 wt% DCP.

Blended cement pastes incorporating DCP generated similar or lower bound water and portlandite content compared to the control paste (I-CTRL). Pastes incorporating DCPs prepared at higher temperatures generated higher bound water compared to other blended pastes, but this did not translate to better strength performance. Incorporation of DCP in the pastes results in high initial heat generation in the first 1–2 h and expedites the acceleration period. However, pastes incorporating DCPs produced at higher temperatures (550 °C/24 h and 750 °C/2 h) show significant retardation of the main hydration of C3S, in line with the lower C3S hydration rate (according to the Rietveld analysis results of in situ XRD) of such pastes at that time.

DCP-blended cements contain higher levels of crystalline ettringite and AFm hydration products throughout the cement hydration process. AFm phases form early, and gypsum is depleted before the main C3S hydration event, indicating undersulfation and disturbance of the regular hydration process, which could affect further hydrate nucleation, growth and microstructure development.

The best strength performance for CEM I pastes was observed for DCI/350 °C/2 h, followed by DCI/550 °C/2 h, 550 °C/24 h and 750 °C/2 h. Therefore, dehydration at temperatures exceeding 350 °C is not recommended for this DC type. For CEM III pastes, DCIII/550 °C/24 h showed the best early age strength performance, while DCIII/350 °C/2 h displayed similar strength activity at later ages (after 28 days). Hence, increasing the dehydration temperature to 550 °C for extended periods is not justifiable unless high early age strength is desired. All DCIII materials and their respective blended mixtures had a slower rate of strength development compared to the corresponding DCI materials and pastes. However, they exhibited higher strength results at later ages compared to DCI.

## Figures and Tables

**Figure 1 materials-17-02659-f001:**
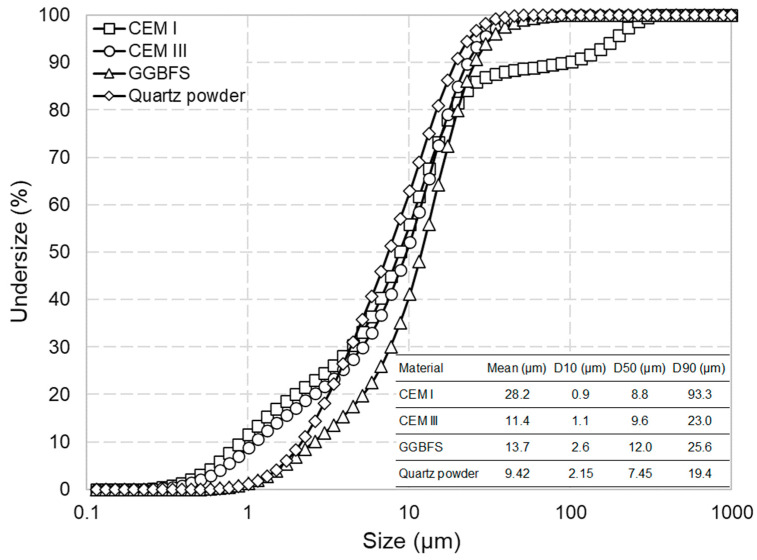
The particle size distribution of the raw materials.

**Figure 2 materials-17-02659-f002:**
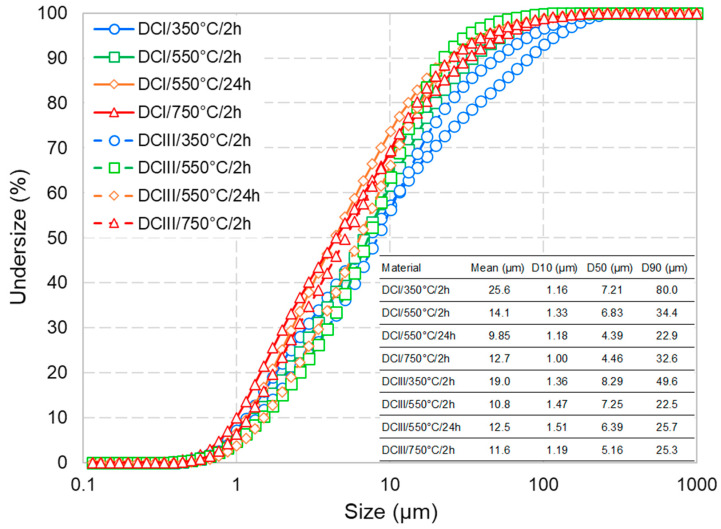
The particle size distribution of the dehydrated cement paste powders treated under different conditions.

**Figure 3 materials-17-02659-f003:**
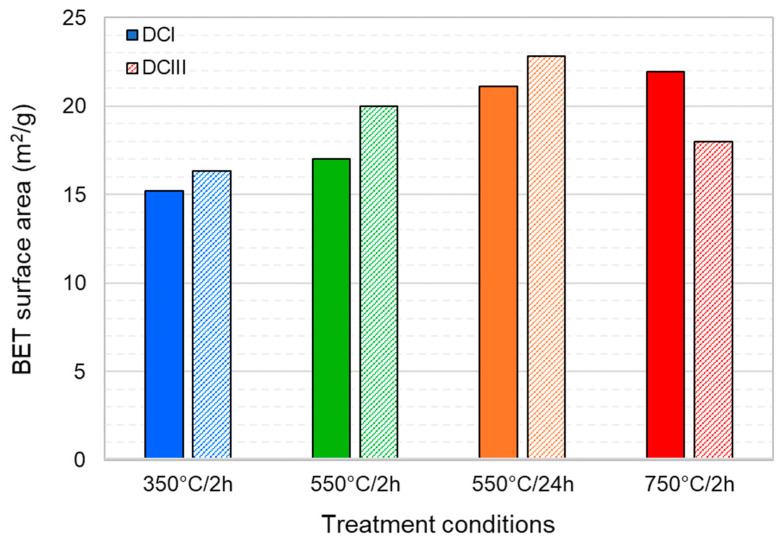
BET surface areas of the dehydrated cement paste powders treated under different conditions. DCI are presented with solid fill colors while DCIII with pattern fill colors.

**Figure 4 materials-17-02659-f004:**
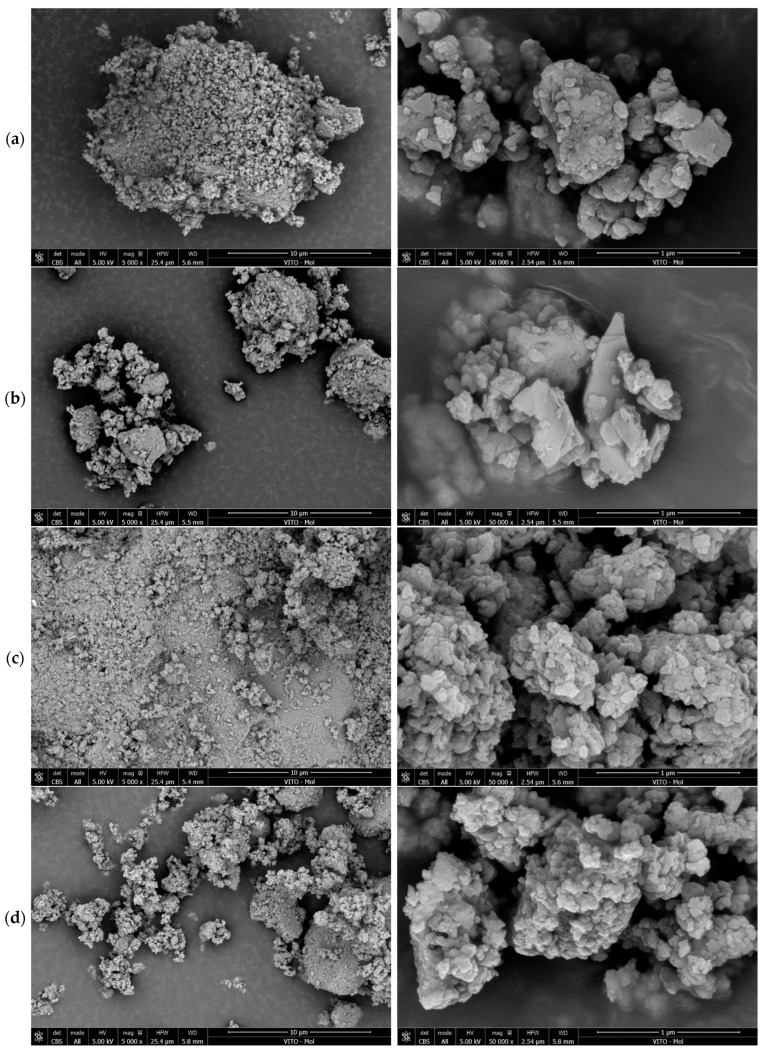
SEM images of the DCI dehydrated cement paste powders at 5000× (**left**) and 50,000× (**right**) magnifications. (**a**) 350 °C/2 h, (**b**) 550 °C/2 h, (**c**) 550 °C/24 h, (**d**) 750 °C/2 h.

**Figure 5 materials-17-02659-f005:**
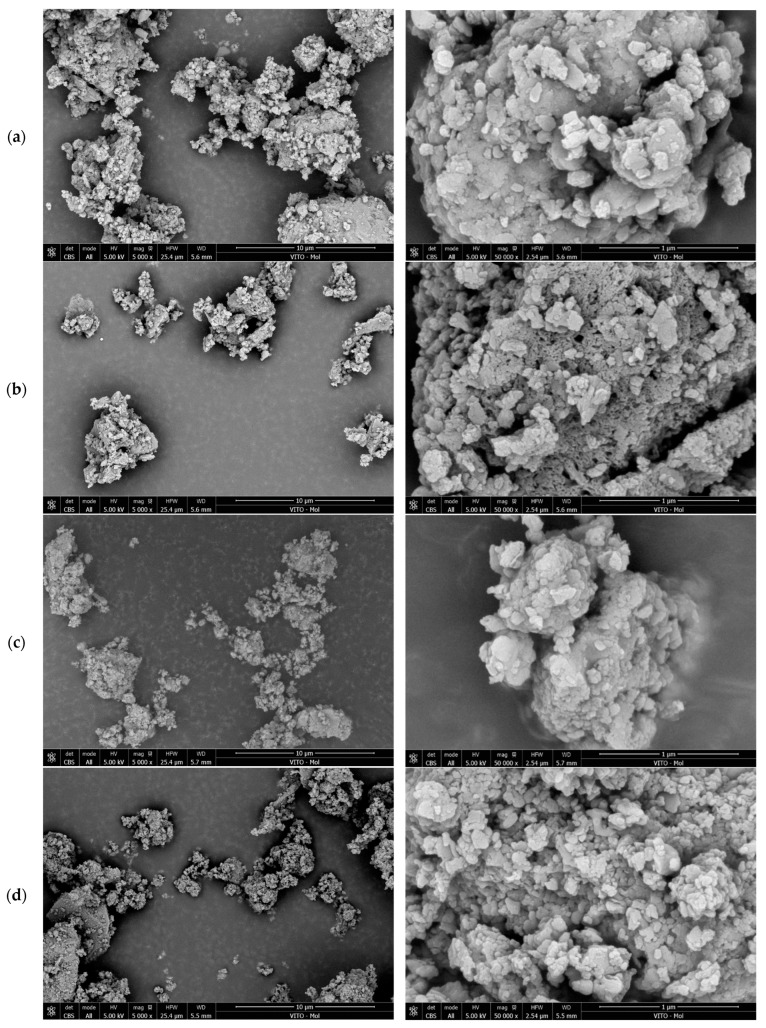
SEM images of the DCIII dehydrated cement paste powders at 5000× (**left**) and 50,000× (**right**) magnifications. (**a**) 350 °C/2 h, (**b**) 550 °C/2 h, (**c**) 550 °C/24 h, (**d**) 750 °C/2 h.

**Figure 6 materials-17-02659-f006:**
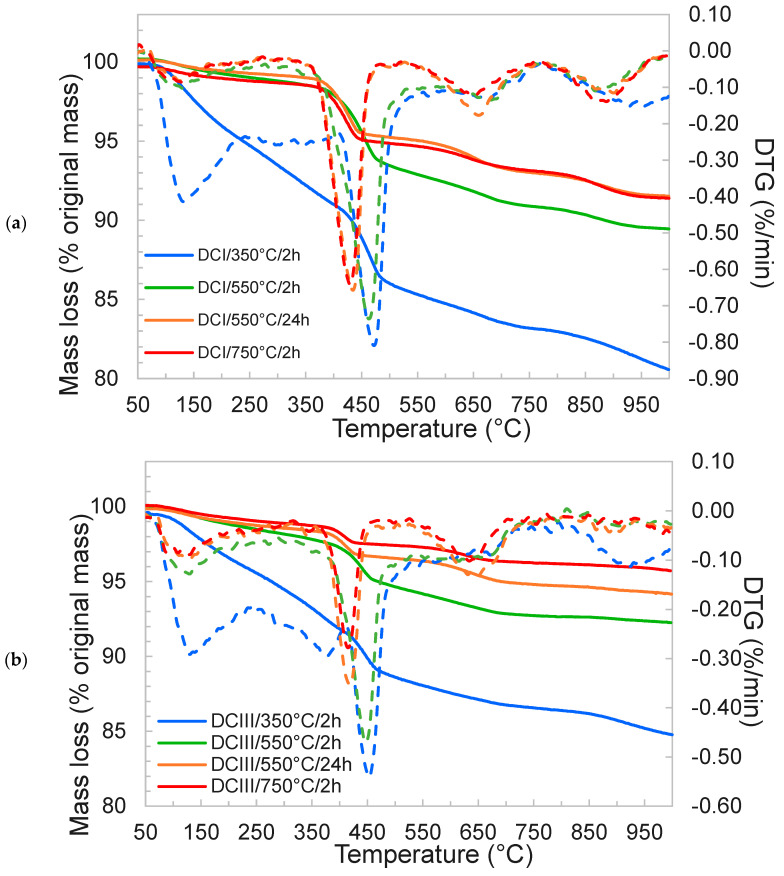
TGA and DTG plots of the dehydrated cement paste powders. (**a**) DCI, (**b**) DCIII. Solid lines present the mass loss and dashed lines the DTG.

**Figure 7 materials-17-02659-f007:**
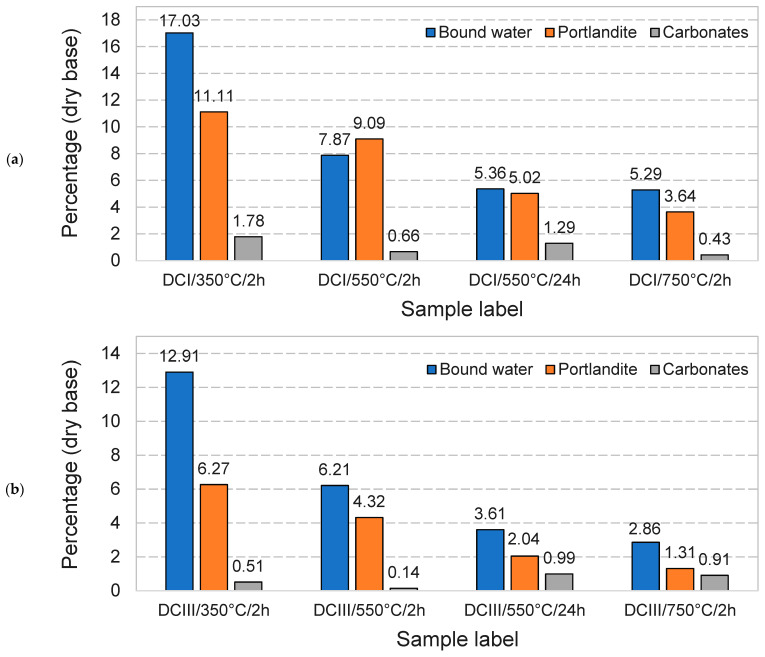
Bound water, portlandite and carbonate contents of the dehydrated cement paste powders. (**a**) DCI, (**b**) DCIII.

**Figure 8 materials-17-02659-f008:**
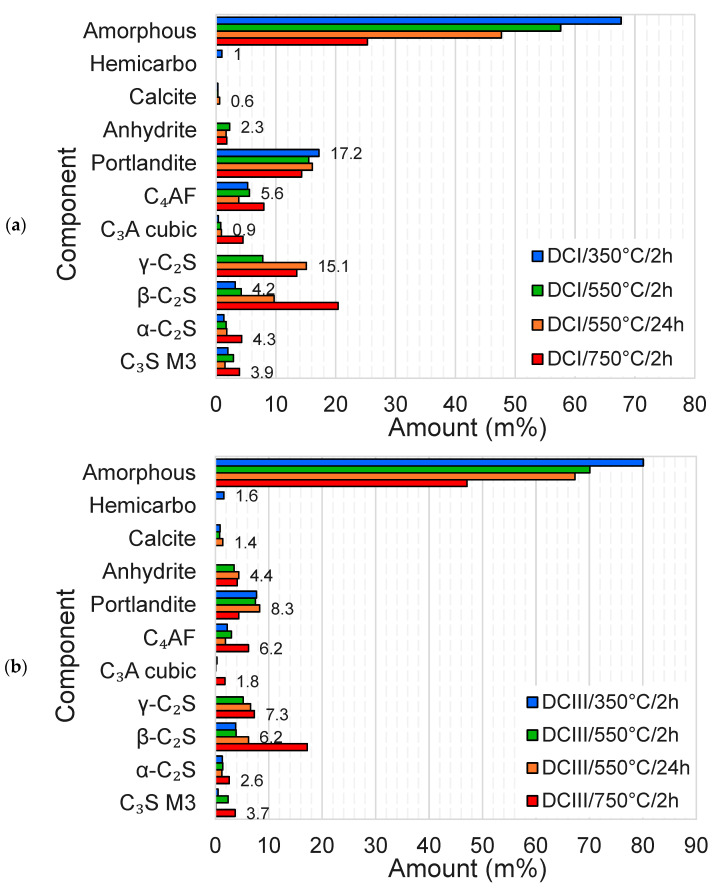
Rietveld analysis results of the dehydrated cement paste powders. (**a**) DCI, (**b**) DCIII.

**Figure 9 materials-17-02659-f009:**
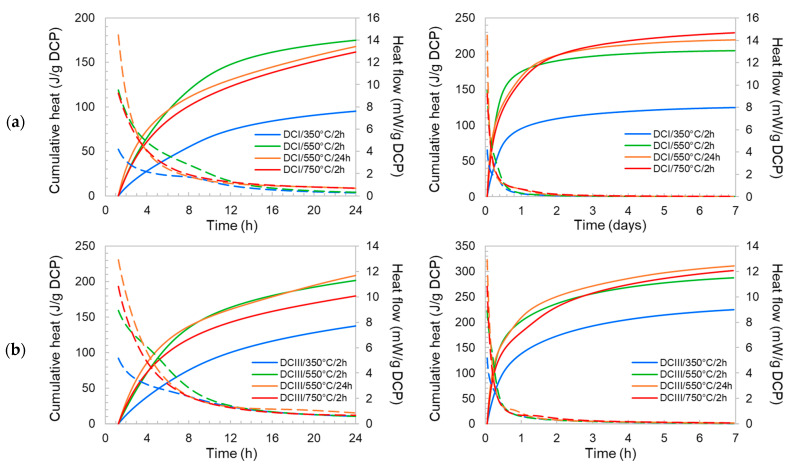
The R3 cumulative heat release and specific heat flow plots of dehydrated cement paste powders in the first 24 h (**left**) and 7 days (**right**). (**a**) DCI, (**b**) DCIII. The solid lines present the cumulative heat and the dashed lines the heat flow.

**Figure 10 materials-17-02659-f010:**
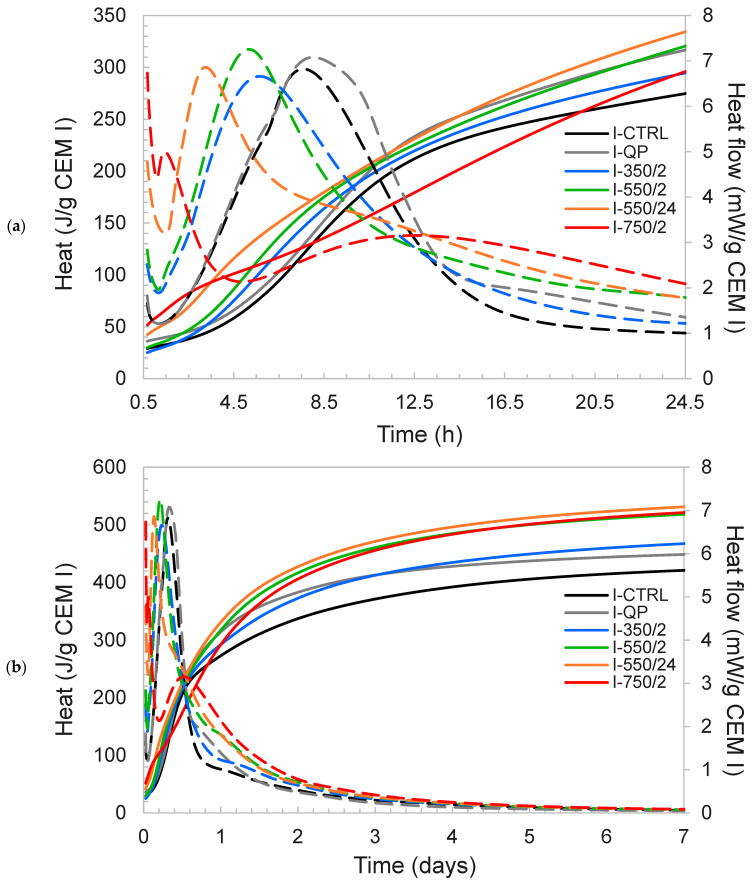
The isothermal calorimetry results of CEM I pastes (**a**) up to 24 h and (**b**) up to 7 d. The heat is presented with solid lines while the heat flow with dashed lines.

**Figure 11 materials-17-02659-f011:**
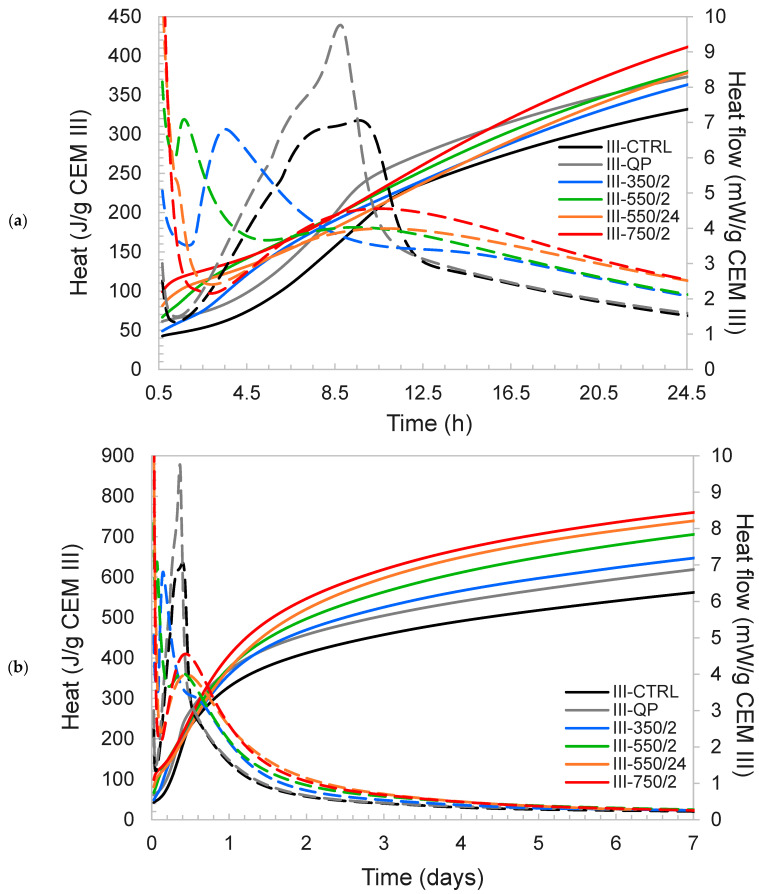
The isothermal calorimetry results of CEM III pastes (**a**) up to 24 h and (**b**) up to 7 d. The Heat is represented with solid lines and Heat flow with dashed lines.

**Figure 12 materials-17-02659-f012:**
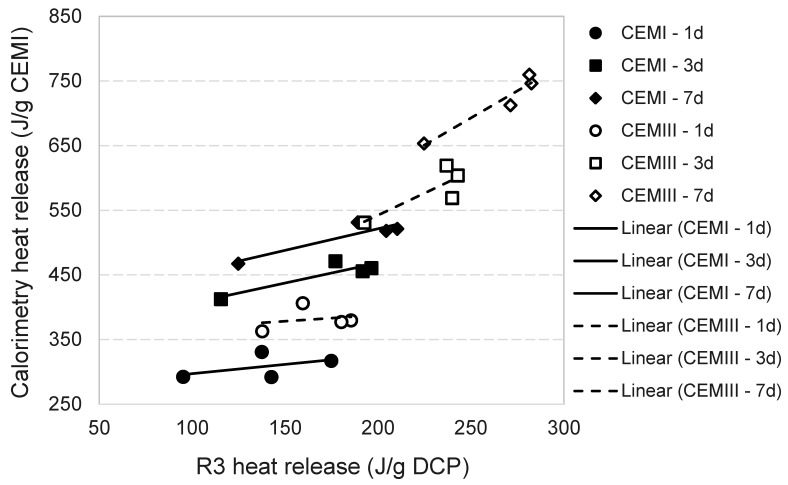
The isothermal calorimetry versus R3 heat release results for CEM I and CEM III pastes at different ages (the embedded table shows the correlation coefficients between calorimetry and R3 heat release and corresponding *p*-values at different test durations).

**Figure 13 materials-17-02659-f013:**
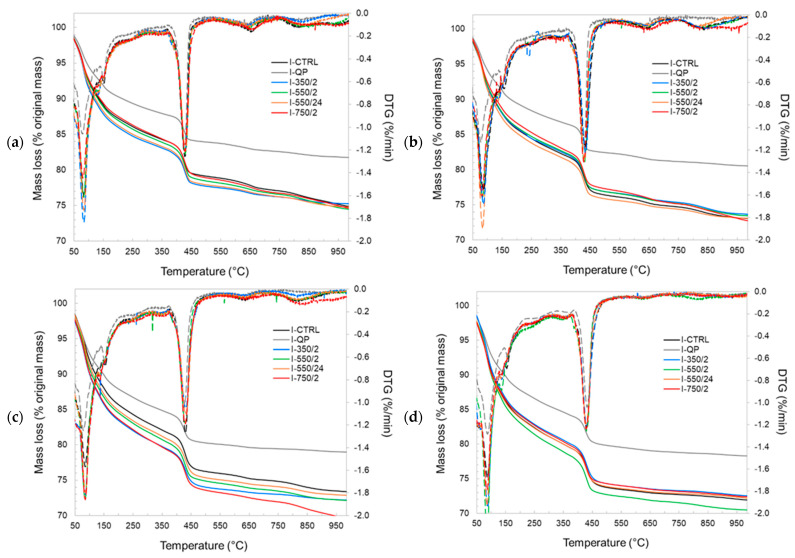
The TGA and DTG curves of the CEM I pastes. (**a**) 3 d, (**b**) 7 d, (**c**) 28 d, (**d**) 91 d. The mass loss is presented with solid lines and the DTG with dashed lines.

**Figure 14 materials-17-02659-f014:**
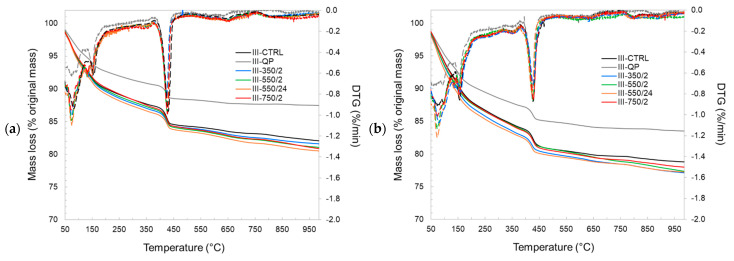
The TGA and DTG curves of the CEM III pastes. (**a**) 3 d, (**b**) 7 d, (**c**) 28 d, (**d**) 91 d. The mass loss is presented with solid lines and the DTG with dashed lines.

**Figure 15 materials-17-02659-f015:**
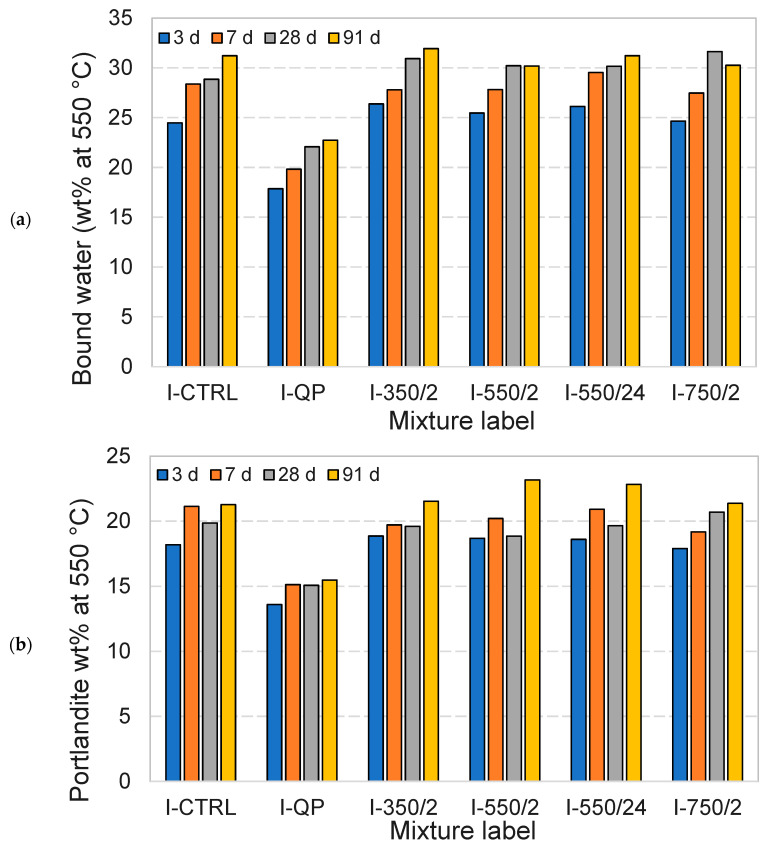
The bound water (**a**) and portlandite content (**b**) values of the CEM I pastes (values normalized to the binder content).

**Figure 16 materials-17-02659-f016:**
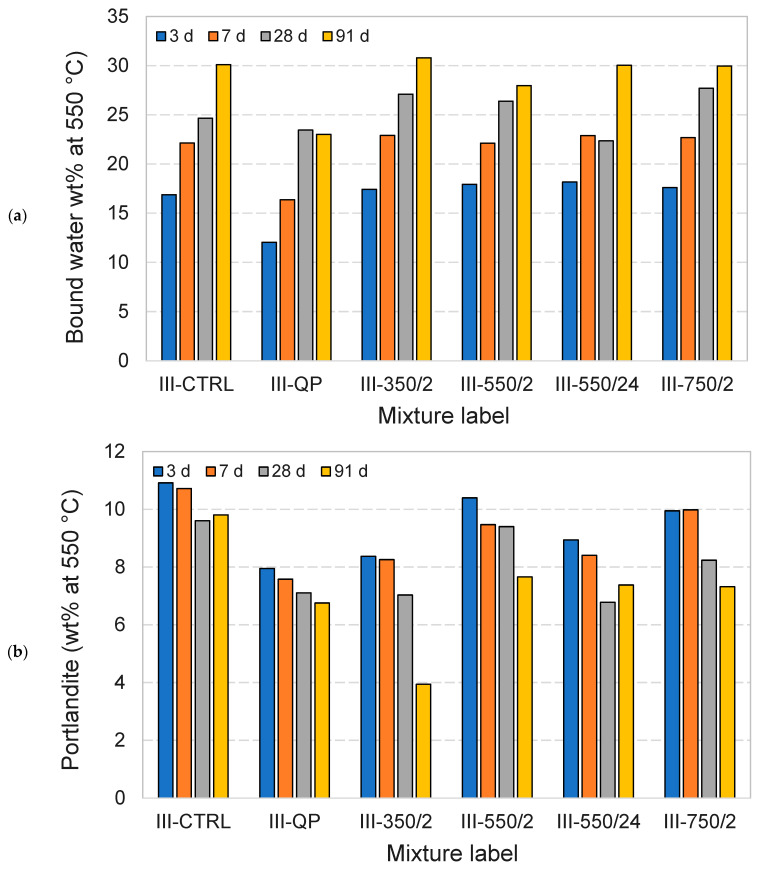
The bound water (**a**) and portlandite content (**b**) values of the CEM III pastes (values normalised to the binder content).

**Figure 17 materials-17-02659-f017:**
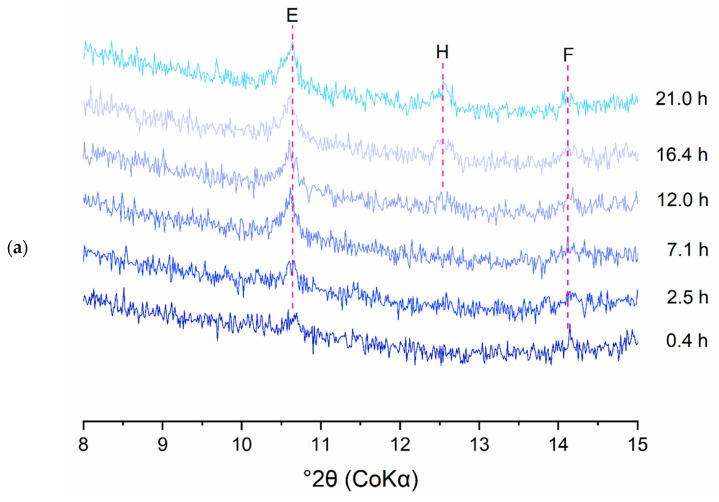
The in situ XRD pattern of I-350/2 paste over time. (**a**) 8–15°, (**b**) 33–41°; A: alite, B: belite, E: ettringite, F: ferrite, H: hemicarboaluminates, P: portlandite.

**Figure 18 materials-17-02659-f018:**
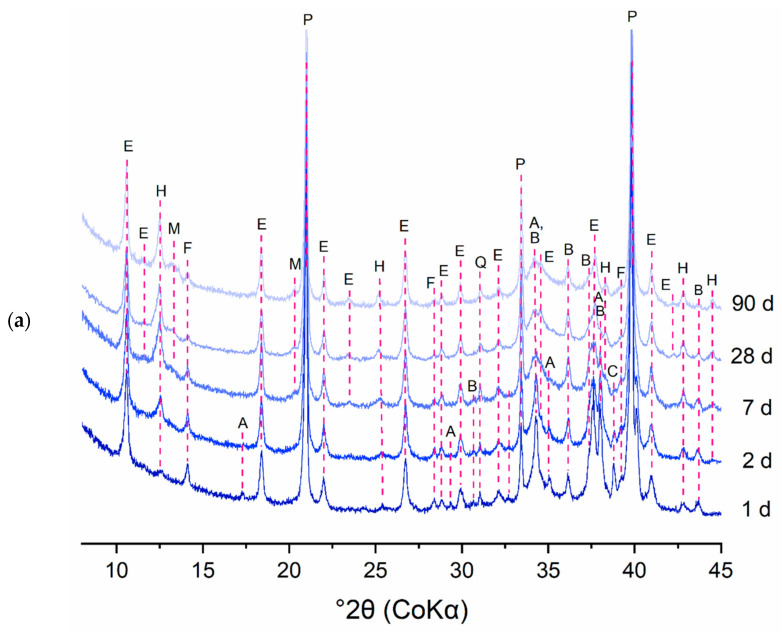
The ex situ XRD pattern of (**a**) I-350/2 paste and (**b**) III-350/2 pastes over time; A: alite, AF: AFm phases, B: belite, C: C_3_A; E: ettringite, F: ferrite, H: hemicarboaluminates, M: monocarboaluminates; P: portlandite, Q: quartz, T: hydrotalcite.

**Figure 19 materials-17-02659-f019:**
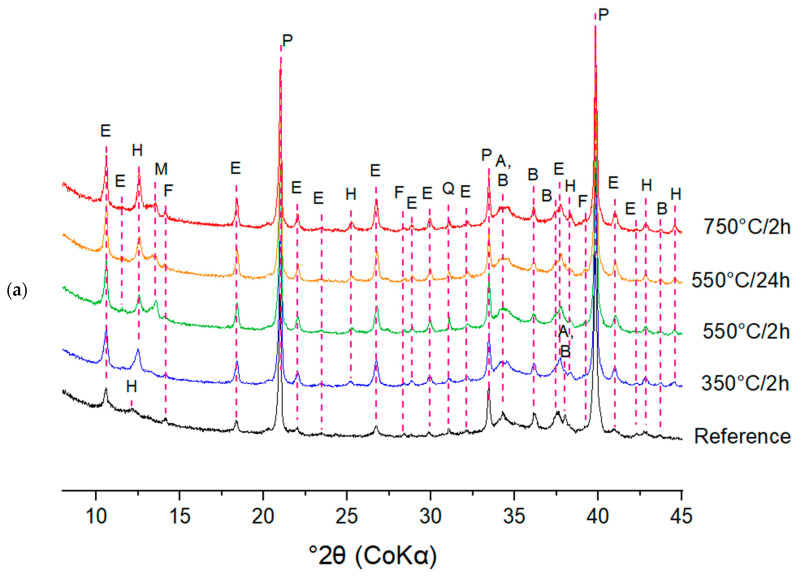
The ex situ XRD patterns of (**a**) CEM I series and (**b**) CEM III series at 28 d; A: alite, AF: AFm phases, B: belite, E: ettringite, F: ferrite, H: hemicarboaluminates, M: monocarboaluminates; P: portlandite, Q: quartz, T: hydrotalcite.

**Figure 20 materials-17-02659-f020:**
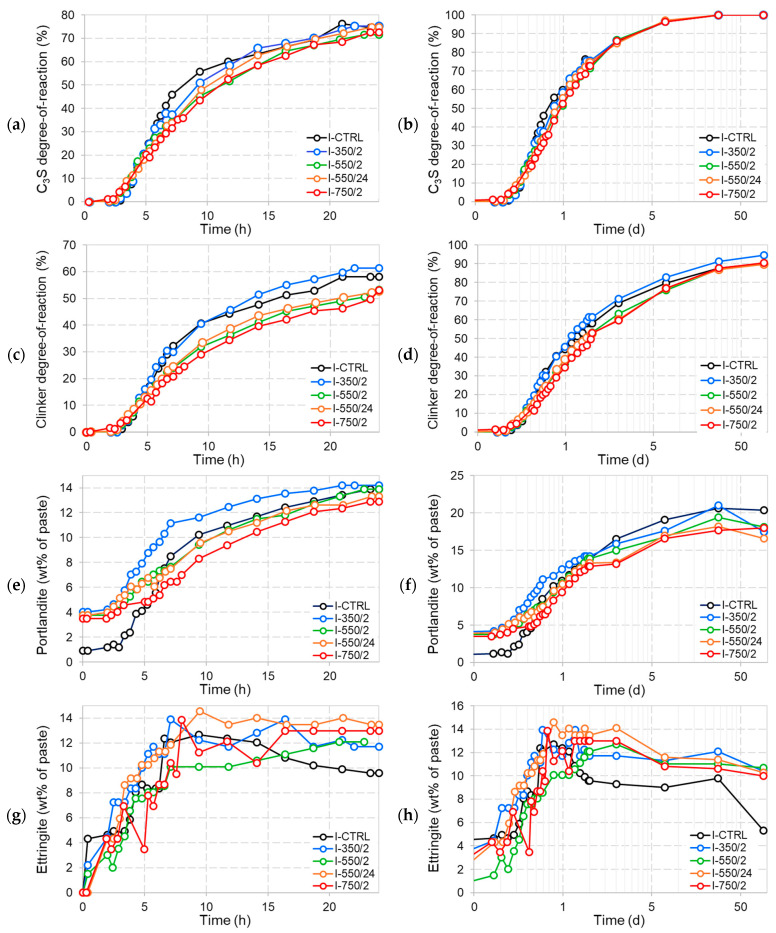
The evolution of primary phases in the CEM I pastes over time (**left**: first 24 h; in situ, **right**: up to 91 d; ex situ). (**a**,**b**) C_3_S degree-of-reaction, (**c**,**d**) clinker degree-of-reaction, (**e**,**f**) portlandite content, (**g**,**h**) ettringite.

**Figure 21 materials-17-02659-f021:**
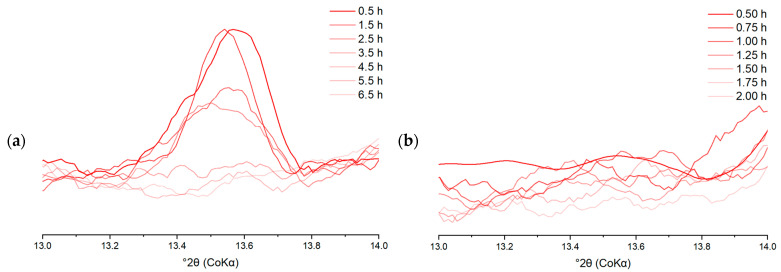
The evolution of gypsum (primary peak) in the in situ XRD diffractograms. (**a**) I-CTRL, (**b**) I-350/2.

**Figure 22 materials-17-02659-f022:**
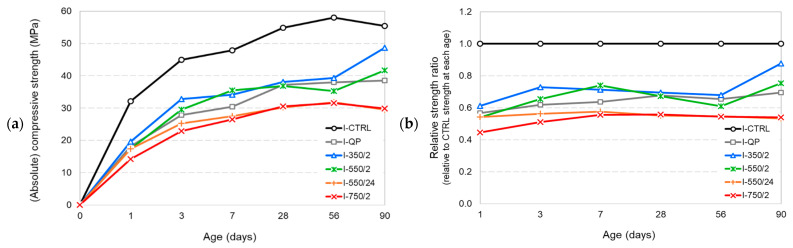
The compressive strength results of the CEM I mortars. (**a**) Absolute compressive strength results, (**b**) relative strength ratios with respect to the strength of the control mixtures at the same age.

**Figure 23 materials-17-02659-f023:**
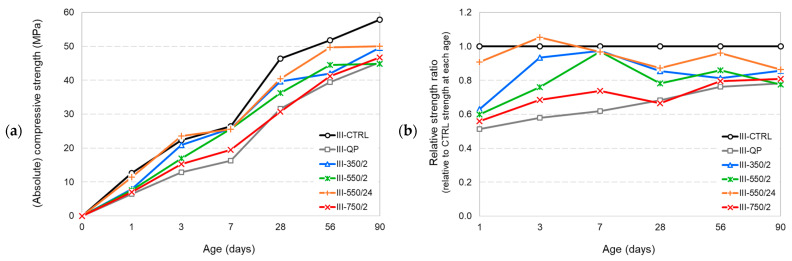
The compressive strength results of the CEM III mortars. (**a**) absolute compressive strength results, (**b**) relative strength ratios with respect to the strength of the control mixtures at the same age.

**Figure 24 materials-17-02659-f024:**
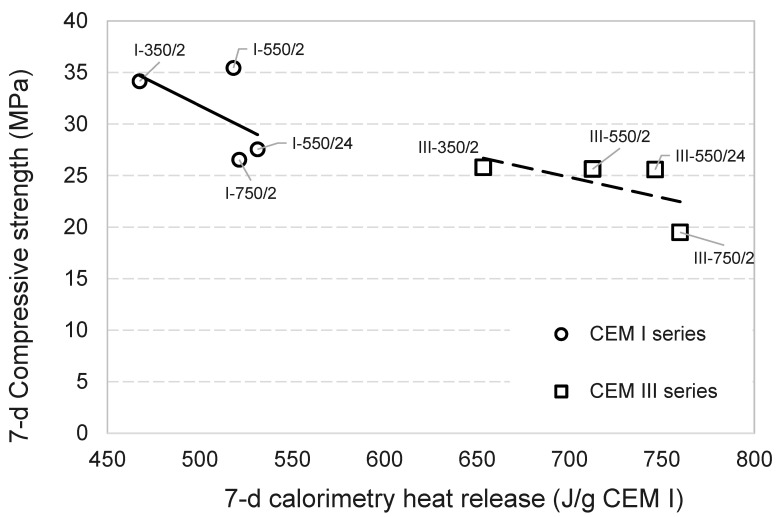
The 7-d compressive strength of blended CEM I and CEM III pastes versus the 7-d calorimetry heat release.

**Table 1 materials-17-02659-t001:** The chemical composition, physical properties and mineral composition of the binder materials. * SSA: Specific Surface Area.

Quantity	Content (m%)		
	CEM I	GGBFS	CEM III
CaO	60.7	37.5	51.8
SiO_2_	18.5	34.5	24.3
Al_2_O_3_	4.89	14.4	8.11
MgO	1.63	7.84	4.19
Fe_2_O_3_	2.88	0.40	1.85
TiO_2_	0.31	1.29	0.55
MnO	0.05	0.31	0.18
SO_3_	2.53	1.23	2.00
K_2_O	0.84	0.74	0.74
Na_2_O	0.12	0.47	0.22
Na_2_O_eq_	0.67	0.96	0.71
LOI (%)	1.47	0.97	1.95
C_3_S M3	59.5	–	33.8
β-C_2_S	17.1	–	9.1
C_3_A cubic	5.2	–	2.8
C_4_AF	10.0	–	6.1
Portlandite	1.3	–	0.1
Gypsum	1.6	–	0.8
Anhydrite	0.9	–	0.2
Basanite	3.1	–	1.6
Quartz	0.6	–	0.17
Periclase	0.7	–	0.61
Amorphous	0.0	100	44.7
SSA (m^2^/g) *	1.76	1.35	1.42

**Table 2 materials-17-02659-t002:** Overview of the studied dehydrated cement paste samples.

Starting Material	Long Name	Dehydration Temperature (°C)	Dwelling Time (h)
CEM I	DCI/350 °C/2 h	350	2
	DCI/550 °C/2 h	550	2
	DCI/550 °C/24 h	550	24
	DCI/750 °C/2 h	750	2
CEM III	DCIII/350 °C/2 h	350	2
	DCIII/550 °C/2 h	550	2
	DCIII/550 °C/24 h	550	24
	DCIII/750 °C/2 h	750	2

**Table 3 materials-17-02659-t003:** The R3 cumulative heat results for the dehydrated cement paste powders.

DCP Label	1 Day	3 Days	7 Days
	J/g SCM	J/g SCM	J/g SCM
DCI/350 °C/2 h	95	115	125
DCI/550 °C/2 h	175	196	204
DCI/550 °C/24 h	137	177	189
DCI/750 °C/2 h	143	192	210
DCIII/350 °C/2 h	138	193	225
DCIII/550 °C/2 h	185	240	271
DCIII/550 °C/24 h	180	243	283
DCIII/750 °C/2 h	160	237	281

**Table 4 materials-17-02659-t004:** The correlation coefficients (and corresponding *p*-values) between the amount of different phases and the 7-d R3 cumulative heat release values.

DC Type	R3 (Duration)	C_3_S M3	α-C_2_S	β-C_2_S	γ-C_2_S	C_3_A Cubic	C_4_AF	Portlandite	Anhydrite	Amorphous
DCI	1 d	0.42 (0.58)	0.22 (0.78)	0.16 (0.84)	0.54 (0.46)	0.19 (0.81)	0.15 (0.85)	−0.66 (0.34)	0.97 (0.03)	−0.33 (0.67)
	3 d	0.52 (0.48)	0.51 (0.49)	0.5 (0.5)	0.78 (0.22)	0.48 (0.52)	0.29 (0.71)	−0.85 (0.15)	0.99 (0.01)	−0.65 (0.35)
	7 d	0.56 (0.44)	0.6 (0.4)	0.59 (0.41)	0.82 (0.18)	0.57 (0.43)	0.35 (0.65)	−0.89 (0.11)	0.96 (0.04)	−0.73 (0.27)
DCIII	1 d	0.11 (0.89)	−0.18 (0.82)	−0.11 (0.89)	0.71 (0.29)	−0.32 (0.68)	−0.1 (0.9)	0.22 (0.78)	0.81 (0.19)	−0.2 (0.8)
	3 d	0.37 (0.63)	0.24 (0.76)	0.34 (0.66)	0.95 (0.05)	0.11 (0.89)	0.29 (0.71)	−0.18 (0.82)	0.98 (0.02)	−0.61 (0.39)
	7 d	0.42 (0.58)	0.38 (0.62)	0.5 (0.5)	0.99 (0.01)	0.26 (0.74)	0.41 (0.59)	−0.31 (0.69)	1 (0)	−0.73 (0.27)

## Data Availability

Data are contained within the article.

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
