# Peer review of "Thermal Reactivation of Hydrated Cement Paste: Properties and Impact on Cement Hydration"

_materials, 2024, doi:10.3390/ma17112659_

Round 1
Reviewer 1 Report
Comments and Suggestions for Authors
The manuscript complies with the aims and scope of “Materials, MDPI. It comprises an exemplary study of a large set of different curing regimes of different DCP based on different cement types and with a very analytical and methodological set of characterization methods. Minor clarifications are advised.
1.Introdution
Lines 67-73: please mention the characterization techniques by which these compounds were traced (XRD, TGA/dTG and/or SEM-EDS?)
2. Materials and Methods:
Lines 128-129: The background to the selected target dwell temperatures is very clear in the intro (lines 74-82). Please mention why in particular you selected 550oC to be heated for 24h in addition to the 2h target dwell temperature?
3.Results and Discussion
Line 292: please correct: Figure 5b (not 5d)
Lines 334-335: please correct
Lines 477-479: please mention the exact process in materials and methods
Figure 17a,b: according to materials and methods you took 1 diffractographs every 15 minutes for 24 h. What selection criterion did you apply for the six specific ones displayed in these figures?
Line 633: “ even better than the control mortar at early ages (< 3 d)”: please add and correct: I even better than the control mortar II-QP at early ages (<7 d)”
Reviewer 2 Report
Comments and Suggestions for Authors
The study investigated the performance of thermo-activated cement pastes, showing that specific heat treatments affected the reactivity and mechanical strength of blended cements.
The paper is well written, the methods section is sufficiently detailed, the presentation of the results is brilliant, condensing a lot of information in the figures without making them difficult to follow. The paper could be improved in two aspects which, in my opinion, should not be considered as major corrections but as minor ones:
1) I think the authors should state a hypothesis (or an answer to their research question that is a priori consistent with their aim). This would make the paper easier to follow. The authors have set out the variables they are trying to use to demonstrate their aim, but I still think it is better to use a clear hypothesis.
2) The conclusions listed are usually not to my liking, but by no means should this comment be taken negatively, it is my point of view. But what I would like is for the authors to say at the beginning of the conclusions whether or not their hypothesis has been fulfilled. Then they can conclude whether or not their variables have allowed them to prove that hypothesis.
Other minor formatting and typographical corrections are detailed in the attached PDF file, which I hope the authors will find useful.
